



# Atmospheric impacts of chlorinated very short-lived substances over the recent past. Part 1: the role of transport

Ewa M. Bednarz[1,a], Ryan Hossaini[1,2], Martyn P. Chipperfield[3,4], N. Luke Abraham[5,6], and Peter Braesicke[7]

1. Lancaster Environment Centre, Lancaster University, Lancaster, UK
2. Centre of Excellence in Environmental Data Science, Lancaster University, Lancaster UK.
3. School of Earth and Environment, University of Leeds, Leeds, UK
4. National Centre for Earth Observation (NCEO), University of Leeds, Leeds, UK
5. Department of Chemistry, University of Cambridge, Cambridge, UK
6. National Centre for Atmospheric Science (NCAS), UK
7. Karlsruhe Institute of Technology, Karlsruhe, Germany
a. now at: Sibley School of Mechanical and Aerospace Engineering, Cornell University, Ithaca, NY, USA

*Correspondence to*: Ewa M. Bednarz (ewa.bednarz@cornell.edu)

**Abstract.**

Atmospheric impacts of chlorinated very short-lived substances (Cl-VSLS) over the first two decades of the 21st century are assessed using the UM-UKCA chemistry-climate model; this constitutes the most up-to-date assessment as well as the first study to simulate Cl-VSLS impacts using a whole atmosphere chemistry-climate model. We examine the Cl-VSLS responses using a small ensemble of free-running simulations as well as two pairs of integrations where the meteorology was 'nudged' to either ERA5 or ERA-Interim reanalysis.

The stratospheric chlorine source gas injection due to Cl-VSLS estimated from the free-running integrations doubled from ~40 ppt Cl in 2000 to ~80 ppt Cl in 2019. Combined with chlorine product gas injection, the integrations showed ~130 ppt of total Cl reaching the stratosphere in 2019 due to Cl-VSLS. The use of the nudged model significantly increased the abundance of Cl-VSLS simulated in the lower stratosphere relative to the free-running model. Averaged over 2010-2018, simulations nudged to ERAI-Interim and ERA5 showed up to ~20 ppt (i.e. a factor of two) and up to ~10 ppt (i.e. ~50%), respectively, more Cl in the lower stratosphere in the form of Cl-VSLS source gases compared to the free-running case. These differences can be explained by the corresponding differences in the speed of the large-scale circulation. The results illustrate the strong dependence of the simulated stratospheric Cl-VSLS levels to the choice between free-running versus nudged set-up, and to the reanalysis dataset used for nudging.

Temporal changes in Cl-VSLS are found to have significantly impacted recent HCl and $COCl_2$ trends in the model. In the tropical lower stratosphere, the inclusion of Cl-VSLS reduced the magnitude of the negative HCl and $COCl_2$ trends (e.g. from ~-8 %(HCl)/decade and ~ -4 ppt($COCl_2$)/decade at ~20 km to ~-6 %(HCl)/decade and ~ 2 ppt($COCl_2$)/decade in the





free running simulations) and gave rise to positive tropospheric trends in both tracers. In the tropics, both the free-running and nudged integrations with Cl-VSLS included compared much better to the observed trends from ACE-FTS than the analogous simulations without Cl-VSLS. Since observed HCl trends provide information on the evolution of total stratospheric chlorine and, thus, the effectiveness of the Montreal Protocol, our results demonstrate that Cl-VSLS are a

confounding factor in the interpretation of such data and should be factored into future analysis. Unlike the nudged model runs, the ensemble mean free-running integrations did not reproduce the hemispheric asymmetry in the observed mid-latitude HCl and $COCl_2$ trends related to short-term dynamical variability. The individual ensemble members also showed a considerable spread of the diagnosed tracer trends, illustrating the role of natural interannual variability in modulating the diagnosed responses, and the need for caution when interpreting both model and observed tracer trends derived over a

relatively short time period.

## 1. Introduction

Stratospheric ozone plays a crucial role in shielding the Earth's surface from harmful UV radiation. Its absorption of shortwave radiation also controls stratospheric temperatures, which in turn can feed back on atmospheric transport and influence surface climate (e.g. Nowack et al., 2015). When in the troposphere and lower stratosphere, ozone also absorbs

longwave radiation, thereby contributing to the greenhouse effect. The important role of halogens in controlling stratospheric ozone levels is now well understood (e.g. WMO, 2018). So far, most attention has been given to long-lived chlorinated and brominated ozone depleting substances (ODSs), of which significant past emissions have been the principal cause of stratospheric ozone depletion, with subsequent impacts on levels of surface UV (e.g. Bais et al., 2018) and surface climate (e.g. Son et al. 2009).


Another class of halogenated compounds that are also thought to contribute to stratospheric ozone depletion are Very Short-Lived Substances (VSLS). These gases have atmospheric lifetimes near the surface of less than about 6 months and, hence, are not well mixed in the troposphere (e.g. Engel et al., 2018). In the past two decades, a wealth of research has examined the sources and sinks of brominated VSLS (e.g. bromoform, $CHBr_3$) of predominantly oceanic origin (e.g. Sturges et al., 2000;

Quack and Wallace, 2003), along with their contribution to stratospheric bromine and impact on ozone (e.g. Hossaini et al., 2016a; Oman et al., 2016; Wales et al., 2018, Keber et al., 2020). However, few studies have considered the atmospheric impacts of chlorinated VSLS (Cl-VSLS), which have significant anthropogenic sources and include dichloromethane ($CH_2Cl_2$), chloroform ($CHCl_3$), perchloroethylene ($C_2Cl_4$) and ethylene dichloride ($C_2H_4Cl_2$), among others. There is strong evidence that Cl-VSLS can enter the stratosphere, thereby affecting stratospheric ozone, based on measurements of the

above compounds and their products in the upper troposphere/lower stratosphere region (UTLS) and modelling (e.g. Laube et al., 2008; Leedham Elvidge et al., 2015; Oram et al., 2017; Harrison et al., 2019; Hossaini et al., 2019).



The sources of Cl-VSLS were recently reviewed by Chipperfield et al. (2020). Briefly, $CH_2Cl_2$ is used in a range of emissive applications (e.g. as a solvent for paint stripping) and as a feedstock in the production of HFC-32 (Feng et al., 2018). Global

total $CH_2Cl_2$ emissions of ~1 Tg/yr have been inferred for 2017 from inversion analysis of atmospheric observations, with Asian emissions increasing markedly over the recent period (Feng et al., 2018; Claxton et al., 2020). For $CHCl_3$, its industrial use is principally as a feedstock for HCFC-22 production, which itself has a range of both emissive and non-emissive uses. Global total $CHCl_3$ emissions of 324 Gg/yr in 2015 were inferred from inverse modelling, with emissions from China suggested as the dominant source location (Fang et al., 2019).


Past and future changes in stratospheric ozone continue to be the subject of multi-model assessment studies (e.g. Dhomse et al., 2018) employing chemistry-climate models (CCMs). However, none of the CCMs participating in such studies, to our knowledge, has so far included Cl-VSLS, thereby omitting a process of potential direct importance to the simulated ozone fields. Past studies examining the atmospheric impacts of Cl-VSLS have employed offline stratospheric chemistry-transport

models (CTM) driven by meteorological reanalysis (e.g. Hossaini et al., 2015; 2019, Claxton et al., 2020). While being able to accurately represent the background interannual dynamical variability, the use of prescribed meteorology meant those studies were unable to simulate the radiative feedback of ozone, and ozone changes, on the underlying stratospheric temperatures and circulation.

In this study we assess the atmospheric impacts of Cl-VSLS using the Met Office's Unified Model coupled to the United Kingdom Chemistry and Aerosol (UM-UKCA) CCM. The study is divided into three parts, constituting the most up-to-date end-to-end assessment of the Cl-VSLS impacts. Part 1 here assesses the atmospheric signatures of Cl-VSLS using a small ensemble of free-running integrations over 2000-2019.. The focus is  on quantifying the stratospheric input of Cl-VSLS and its impacts on HCl and $COCl_2$ trends; the latter are important proxies for inferring changes in total stratospheric chlorine

content and, thus, monitoring the effectiveness of the Montreal Protocol. We complement these free-running simulations with integrations where the model meteorology was 'nudged' towards the observed conditions, as given by two different reanalysis datasets. This comparison between the free-running and nudged approaches provides insight into and demonstrates the role of atmospheric dynamics in modulating the atmospheric responses to Cl-VSLS. It also demonstrates the importance of the choice of model set-up for the inferred responses.


Section 2 of this paper describes the UM-UKCA model and the simulations performed. Section 3 quantifies the stratospheric input of Cl-VSLS and the resulting impacts on stratospheric chlorine species. Section 4 focuses on the impacts of Cl-VSLS on the recent HCl trends and Section 5 on the recent $COCl_2$ trends. Our main results are summarised and discussed in Section 6.

## 2. Methods

### 2.1. Model description

We use version 11.0 of the UM-UKCA CCM, the atmosphere-only configuration of the UKESM1 Earth System Model (Sellar, et al., 2019). The full description of the model can be found in Sellar et al. (2019) and Archibald et al. (2020). Briefly, the model consists of the Global Atmosphere 7.1 configuration of the version 3 of the Hadley Centre Global Environment Model (GA7.1 HadGEM3, Walters et al., 2019) coupled to the UKCA chemistry and aerosol module (Morgenstern et al., 2009). The horizontal resolution is 1.875° longitude x 1.75° latitude, with 85 vertical levels up to ~84 km on terrain-following hybrid height coordinate.

We use a newly developed 'Double Extended Stratospheric Tropospheric Scheme' (DEST, Bednarz et al., in prep.). The scheme constitutes an extension of the standard StratTrop scheme described in Archibald et al. (2020). It includes an explicit treatment of 14 long-lived ODSs as well as some of the most important chlorinated and brominated VSLS, i.e.: time-varying $CH_2Cl_2$, $CHCl_3$, $C_2Cl_4$ and $C_2H_4Cl_2$ tracers forced at the surface using lower boundary conditions (LBCs, see below) and climatological emissions of the predominately natural VSLS $CHBr_3$, $CH_2Br_2$ $CH_2BrCl$, $CHBr_2Cl$ and $CHBrCl_2$ following Ordonez et al., (2012). A detailed description and evaluation of the scheme can be found in Bednarz et al. (in prep.).

### 2.2. Model experiments

A set of UM-UKCA transient integrations was performed over the period 1990-2019. Observed sea-surface temperatures (SSTs) and sea-ice are prescribed in all runs. For the free-running integrations, we use the CMIP6-recommended dataset described in Durack and Taylor (2016) until 2016, and the Reynolds and Smith (1994) dataset, monthly-averaged, thereafter. . In all simulations, surface concentrations of greenhouse gases and the long-lived ODSs follow Meinshausen et al. (2017) from 1990 to 2014 and then the CMIP6 Shared Socioeconomical Pathway SSP2-4.5 scenario (Meinshausen et al., 2020) from 2015 to 2019. Emissions of aerosols and chemical tracers of importance in the troposphere are as in Archibald et al. (2020) and Sellar et al. (2020) until 2014, and then follow SSP2-4.5. Surface area density of stratospheric sulphate aerosols is from Thomason et al. (2018) until 2014 and climatological thereafter. In all experiments (summarised in Table 1), the period from 1990 to 1999 is treated as a model 'spin-up' and so only model output from 2000 onwards is analysed.

A 3-member ensemble of free-running integrations was carried out without Cl-VSLS. We denote this ensemble as 'BASE'. The experiment spans the period from January 1990 to December 2019 inclusive. A second 3-member ensemble of free-running integrations, denoted 'VSLS', is analogous to BASE but included four of the most important Cl-VSLS ($CH_2Cl_2$, $CHCl_3$, $C_2Cl_4$ and $C_2H_4Cl_2$) constrained at the surface using latitude- and time- dependent LBCs. For each Cl-VSLS, the surface LBCs are applied in the model in five latitude bands (90°S-30°S, 30°S-0, 0-30°N, 30°N-60°N and 60°N-90°N) and vary annually. For a given year, the annual LBCs are calculated by averaging surface measurements from available sites



within each of the five latitude bands. Based on our previous work (Hossaini et al., 2019), measurement data from the NOAA global monitoring network were used for $CH_2Cl_2$ and $C_2Cl_4$, while AGAGE network data were used for $CHCl_3$. For $C_2H_4Cl_2$, latitude-dependent LBCs were estimated (see Hossaini et al., 2016b) based on measurements made during the

2009-2011 HIPPO aircraft campaign (Wofsy, 2011). In the 1990-1999 period (i.e. spin-up), Cl-VSLS LBCs are assumed to be the same as for the year 2000. The time evolution of the global mean surface concentration of each Cl-VSLS is shown in Figure 1, and the evolution in each of the five latitude bands is shown in Figure S1 in the Supplementary Material.

The free-running BASE and VSLS model ensembles were complemented with two sets of similar integrations but using the

specified dynamics configuration of UM-UKCA, i.e. in which model meteorology (in particular temperature, zonal and meridional winds) are 'nudged' towards observed conditions (Telford et al., 2008). These follow European Centre for Medium-Range Weather Forecasts (ECMWF) ERA5 reanalysis (Hersbach et al., 2020) in the first pair of integrations, $VSLS_{SD-5}$ and $BASE_{SD-5}$, and the ECMWF ERA-Interim reanalysis (Dee et al., 2013) in the second pair, $VSLS_{SD-I}$ and $BASE_{SD-I}$. The first pair covers the period until the end of March 2020 and the second pair, limited by the length of the ERA-

Interim dataset, up to August 2019 inclusive. For better consistency with nudging frequency, rather that the monthly mean SSTs/sea-ice data used in the free-running integrations, we use the daily mean Reynold and Smith (1994) dataset throughout the simulation period. Comparison of the two datasets shows overall similar evolution of the monthly mean tropical SSTs (Fig. S2), thereby suggesting that any differences in the simulated responses between the free-running and nudged integrations are not primarily the result of the differences in the imposed SSTs. Note that in the subsequent discussion of Cl-

VSLS impacts, for brevity, we refer to the difference between the ensemble mean free-running VSLS and BASE experiments as 'ΔFR'; and to the difference between the nudged $VSLS_{SD-5}$ and $BASE_{SD-5}$ (or $VSLS_{SD-I}$ and $BASE_{SD-I}$) as 'ΔSD-5' (or 'ΔSD-I').

## 3. Stratospheric input of Cl-VSLS and impacts on stratospheric chlorine species

### 3.1. Free-running simulations

The simulated ensemble mean concentrations of different Cl-VSLS (averaged over the final decade of the VSLS runs, i.e. 2010-2019) are shown in Fig. 2. In accord with the latitudinal gradient in the prescribed Cl-VSLS LBCs (Fig. S1), the highest source gas concentrations occur in the Northern Hemisphere (NH) troposphere, decreasing in magnitude with increasing altitude due to atmospheric degradation processes. We model significant amounts of Cl-VSLS near the tropical tropopause, amounting to ~70 ppt Cl at 17 km (Fig. 2e). The stratospheric source gas injection ('SGI') of chlorine from Cl-

VSLS can be approximated based on their simulated concentrations at 17 km and 25°S-25°N, as shown in Fig. 3. We find that the simulated SGI doubled over the first two decades of the 21$^{st}$ century, with ~80 ppt Cl being injected into the stratosphere in the form of Cl-VSLS in 2019, compared to ~40 ppt Cl in 2000 (black line in Fig. 3b). Combined with the additional Cl reaching the stratosphere in the form of product gases (i.e. product gas injection, 'PGI'), e.g. HCl, we find ~130

ppt of extra Cl reaching the stratosphere in 2019 (compared to ~70 ppt in 2000) due to Cl-VSLS alone (purple line in Fig.
3b).

We note that although the individual ensemble members in these free-running experiments have, by definition, different
meteorology, the simulated stratospheric SGI and PGI of chlorine show very similar inter-annual variability (Fig. 3b). Recall
that all three VSLS ensemble members are forced with identical chemical (i.e. Cl-VSLS) and meteorological (i.e. SSTs and
sea-ice) LBCs. Hence, the results suggest that it is the interannual differences in the Cl-VSLS surface mixing ratios and/or
SSTs/sea-ice data rather than the model internal dynamical variability that is the main driver of variability in SGI and PGI on
interannual timescales.

The transport of Cl from Cl-VSLS into the stratosphere adds to the already elevated stratospheric chlorine levels caused by
past emissions of long-lived ODSs. Averaged over the last decade (2010-2019), our simulations show ~100 ppt of additional
chlorine in the lower/mid-stratosphere in the VSLS experiment compared to BASE (Fig. 2f). This accounts for ~3 % of total
stratospheric chlorine for that period.

Figure 4 speciates the modelled difference in chlorine between the experiments VSLS and BASE. Most of the additional Cl
in the former runs is found as HCl, the principal stratospheric chlorine reservoir, and in $ClONO_2$, ClO and $COCl_2$
(phosgene). The latter is an important product of Cl-VSLS oxidation and an atmospheric degradation product of the longer-
lived source gases $CCl_4$ and $CH_3CCl_3$ (e.g. Fu et al., 2007; Harrison et al., 2019). Here, we estimate that up to 8 ppt of the
$COCl_2$ simulated over the last decade in the VSLS experiment is of Cl-VSLS origin, with Cl-VSLS accounting for the
majority of $COCl_2$ found in the troposphere.  The increase in HCl abundance in the mid-/upper stratosphere brought about
from the inclusion of Cl-VSLS in the model reduces the underestimation of HCl in that region compared to satellite data
(Bednarz et al., in prep.).

### 3.2. Free-running vs nudged simulations

We now consider the sensitivity of the modelled stratospheric chlorine SGI and PGI to the choice between the free-running
and nudged meteorology. We find that the use of the nudged model set-up significantly increases the abundance of Cl-VSLS
in the lower stratosphere (Fig. 5a,c) and hence the stratospheric chlorine SGI from Cl-VSLS (Fig. 3a). For example,
averaged over 2010-2018, the $VSLS_{SD-I}$ run shows up to ~20 ppt (compared to ~20 ppt in the free running VSLS, i.e. a factor
of two) more Cl in the lower stratosphere in the form of source gases relative to the free-running VSLS simulation (Fig. 5c).
There is also a strong dependence on the reanalysis used for nudging, with the run nudged to ERA5 ($VSLS_{SD-5}$) showing up
to ~10 ppt (i.e. ~50%) more Cl in the form of source gases in the subtropics than the free-running runs (Fig. 5a), compared to
~20 ppt more Cl in the runs nudged to ERAI-Interim noted above ($VSLS_{SD-I}$ , Fig. 5c).

These larger stratospheric concentrations of Cl-VSLS arise because of a markedly faster large-scale atmospheric circulation simulated in the nudged runs, particularly in the lower stratosphere (Fig. 6). The faster large-scale circulation accelerates the transport of Cl-VSLS across the tropical tropopause and into the lower stratosphere. Chrysanthou, et al. (2019) found that an

acceleration of tropical upwelling in nudged CCMs, compared to their free-running versions, is a common feature across different models. Here, the differences in the simulated age-of-air (AoA) between the nudged and free-running simulations are larger for the run nudged to ERA-Interim than for the run nudged to ERA5. This corresponds to relatively slower Brewer-Dobson Circulation (BDC) in ERA5 compared to ERAI-Interim, in agreement with previous studies (Diallo, et al., 2021; Ploeger et al., 2021). The lager differences in transport between the free-running and ERA-Interim nudged UM-

UKCA simulations are commensurate with larger difference in Cl-VSLS levels in the lower stratosphere.

Chrysanthou et al (2019) analysed transport in CCMs and showed that while nudging does have a large impact on the resolved large-scale residual circulation in these models, it does not necessarily constrain its mean strength or otherwise improves it compared with the free-running models, similar to the conclusions of Orbe et al. (2018). The performance of the

model stratospheric transport can be assessed by comparing the model AoA with that diagnosed from satellite observations of long-lived tracers. Here, Fig. 7 compares UM-UKCA AoA against the same quantity derived from MIPAS observations of SF6 (Stiller et al., 2020). Note that both the model and the observed AoA were normalised by subtracting the values calculated in each case at the tropical tropopause. We find that in the tropics the stratospheric AoA in the run nudged to ERAI-Interim (VSLS$_{SD-I}$) compares more favourably to the MIPAS data than the AoA in the other simulations. However,

this is not the case in the mid- and high latitudes. Therefore, rather than judging unambiguously which model set-up performs better, we highlight the importance this choice between the free-running and nudged configuration, alongside the choice of reanalysis dataset for the latter, has on the diagnosed Cl-VSLS response.

While the stratospheric concentrations of Cl-VSLS (and hence chlorine SGI) simulated in the nudged runs are significantly

larger than in the free-running runs, the corresponding changes in total stratospheric chlorine (Cl$_{tot}$) are slightly smaller for ΔSD-5 and ΔSD-I than for ΔFR (Fig. 5b,d). This corresponds to the relatively smaller increases in the stratospheric concentrations of product gases in the nudged runs (Fig. S3 and Fig. S4). This likely indicates (i) an enhanced transport of chlorine into the lower stratosphere in the form of Cl-VSLS rather than product gases (i.e. a higher SGI:PGI ratio), and (ii) an accelerated removal of inorganic chlorine from the stratosphere under faster large scale circulation.

**4. The Impacts of Cl-VSLS on HCl trends**

**4.1 Ensemble mean free-running simulations**

Figure 8 shows linear trends in the deseasonalised HCl mixing ratios (MAM 2004- SON 2019) calculated from the free-running (ensemble mean) and nudged experiments. The trends are shown with and without Cl-VSLS, and as a function of

latitude and height. The observed HCl trend derived from ACE-FTS (vn3.5-3.6) satellite data (Boone et al., 2013) is also

shown for comparison. In each case, zonal and monthly mean HCl data was first interpolated onto a 10°-latitude grid and seasonally averaged; the resulting timeseries was then deseasonalised, and a simple linear trend was calculated. Note that for the runs nudged to ERA-Interim (VSLS$_{SD-I}$, BASE$_{SD-I}$), the trends are calculated for a slightly shorter time period from MAM 2004 to JJA 2019 inclusive.

In general, stratospheric HCl mixing ratios have been decreasing since near the turn of the century, in line with the phase-out of long-lived ODSs and decreasing total stratospheric chlorine (Froidevaux et al., 2015; Bernath and Fernando, 2018). In accord, our BASE experiment shows a negative HCl trend that maximises in the tropical lower stratosphere at ~ -8 %/decade at ~20 km altitude (Fig. 8b). There is also a small positive HCl trend in BASE in the tropical upper troposphere (up to ~4 %/decade at 10 km); this positive HCl trend is consistent in terms of the sign with the positive, albeit not statistically

significant, tropospheric HCl trend inferred from the ACE-FTS data (Fig. 8g).

We find that the inclusion of Cl-VSLS (Fig. 8a, Fig. 9) decreases the magnitude of the negative stratospheric HCl trend by around 25% in the tropical lower stratosphere, i.e. from approximately -8 %/decade to -6 %/decade at 20 km in BASE and VSLS, respectively (Fig. 9e)., The HCl trend derived from the VSLS run in the tropics agrees better with the ACE-FTS data

in the lower and mid- stratosphere than the HCl trend diagnosed from BASE. In the tropical upper troposphere, the inclusion of Cl-VSLS significantly magnifies the positive HCl trend, which in the VSLS run reaches up to approximately 14 %/decade at 10 km compared to 4 %/decade in BASE.

Notably, the observed ACE-FTS HCl trend displays a strong asymmetry in its horizontal structure in the lower/mid-

stratosphere, with a statistically significant negative trend (up to approximately -10 %/decade) in the SH mid-latitudes and a very weak, non-significant trend in the NH mid-latitudes (Fig. 7g). This asymmetry has been shown to arise due to the corresponding dynamical variability characterising the recent period (e.g. Mahieu et al., 2014), in particular the apparent southward shift of the BDC found in the observational data (e.g. Ploeger et al., 2015; Wargan et al., 2018). Such horizontal pattern in the lower stratospheric HCl trend is not reproduced by either the ensemble mean VSLS or BASE runs. This is

because the use of an ensemble mean of free-running integrations effectively minimises the effects of any short-term dynamical variability, as illustrated by only very small trends in the corresponding ensemble mean AoA in these runs (Fig. 10a,b).

### 4.2. Free-running vs nudged simulations

In contrast to the free-running integrations, the two pairs of nudged simulation (VSLS$_{SD-5}$/BASE$_{SD-5}$ and VSLS$_{SD-I}$/BASE$_{SD-I}$), by construction, show similar interannual dynamical variability to observations. In particular, all display a negative AoA

trend (i.e. younger air) in the SH mid-latitudes and a positive trend (i.e. older air) in the NH mid-latitudes (Fig.10c-f). This is





a consequence of the apparent southward shift of BDC reported from observational data. Since younger air is associated with lower concentrations of HCl (and vice versa for older air), the HCl trends diagnosed from the nudged runs show strong hemispheric asymmetry (Fig. 7c-f), with markedly stronger (i.e. more negative) HCl trend in the SH mid-latitudes and a

weaker (i.e. less negative) HCl trend in the NH mid-latitudes. Such pattern is thus similar to that found in the ACE-FTS data. While the agreement in the diagnosed HCl trends between the runs nudged to ERA5 and ERA-Interim reanalysis is much closer than between the nudged and free-running integrations, there are still some important qualitative differences between the nudged runs. This is particularly true in the SH lower stratosphere, in accord with differences in the associated AoA trends in that region.


As was the case with the free-running integrations, the inclusion of Cl-VSLS improves the agreement between the simulated and ACE-FTS HCl trends in the tropical lower and mid-stratosphere (Fig. 9b). It also improves the agreement in the NH and SH mid-latitudes (Fig. 9a,c), which was not necessarily the case in the free-running integrations due to the role of short term dynamical variability described above.

**4.3. Individual ensemble members**

Lastly, despite the same boundary conditions and forcings, we diagnosed a considerable range of HCl trends, and their horizontal structures, from the individual ensemble members of the free running integrations (right panels in Fig. 9; Fig. 11). For example, the HCl trends in the SH mid-latitude from the individual VSLS ensemble members at ~17 km vary by over a factor of 3 from ~ -10 %/decade in ENS1 to ~ -3 %/decade in ENS3 (Fig. 9l). This illustrates the role of natural interannual

variability in atmospheric circulation in modulating the diagnosed HCl trends (and Cl-VSLS responses). A range of trends in model AoA are found over this ~15-year- period from the individual ensemble members, with the different members displaying opposing positive and negative AoA trends that appear statistically significant (Fig. S5). This reaffirms the need for caution when interpreting both model and observationally derived trends in atmospheric tracers when calculated over such relatively short time periods.

**5. The Impact of Cl-VSLS on COCl$_2$ trends**

This section discusses recent rends in phosgene, an important product of Cl-VSLS oxidation and an atmospheric degradation product of the longer-lived CCl$_4$ and CH$_3$CCl$_3$ (e.g. Fu et al., 2007; Harrison et al., 2019).

**5.1 Ensemble mean free-running simulations**

Figure 12 shows linear trends in deseasonalised COCl$_2$ mixing ratios over 2004-2019, analogous to the HCl trends in Fig. 8.

Consistent with concurrent long-term decline in atmospheric concentrations of CCl$_4$ and CH$_3$CCl$_3$ (WMO, 2018) and total inorganic chlorine, the stratospheric COCl$_2$ levels have decreased over the recent past, as illustrated by the negative COCl$_2$



trend in the ensemble mean BASE diagnosed throughout the lower stratosphere (Fig. 12b). We find that the inclusion of Cl-VSLS significantly decreases the magnitude of this negative stratospheric $COCl_2$ trend, with its maximum amplitude of ~ -5 ppt/decade and ~ -4 ppt/decade for the ensemble mean BASE and VSLS, respectively ( Fig. 12a-b). Averaged over the tropics, this corresponds to a weakening of the trend from ~ -4 ppt/decade to ~ -2ppt/decade at 21 km for BASE and VSLS, respectively (Fig. S7e). In the troposphere, while only very small and negative trend was simulated in the ensemble mean BASE run, the inclusion of Cl-VSLS leads to a small but positive, statistically significant $COCl_2$ trend (~1ppt/decade at 10 km in the tropics). A qualitatively similar positive tropospheric $COCl_2$ trend was also found in the observed ACE-FTS data (Fig. 12g). Averaged over the tropics (30°S-30°N), the $COCl_2$ trend derived from VSLS thus agrees better with the ACE-FTS data throughout the troposphere and lower stratosphere than the $COCl_2$ trend diagnosed from BASE (Fig. S6).

### 5.2. Free-running vs nudged simulations

While the $COCl_2$ trends diagnosed from the ensemble mean of free running integrations are symmetrical in both hemispheres, this is not the case in the nudged runs (Fig. 12c-f). In particular, the $COCl_2$ trends diagnosed from the nudged runs are stronger (i.e. more negative) in the NH tropics and mid-latitudes than it is the case in the free-running integrations; the maximum amplitudes of the trends are also larger. In the SH sub-tropical and mid-latitudes, the $COCl_2$ trends at ~25 km are small but positive. Such horizontal structure is qualitatively similar to that found in the observed ACE-FTS data (Fig. 12g) and is related to the southward shift of the upwelling part of the BDC (Sect. 4.2), which increases $COCl_2$ in the SH sub-tropics and decreases in the NH.

As was the case with the free-running integrations, the inclusion of Cl-VSLS improves the agreement between the model and ACE-FTS $COCl_2$ trend in the tropics (Fig. S6). The agreement also improves in the NH and SH mid-latitudes (Fig. S6), which was not necessarily the case in the free-running integrations due to the influence of short-term dynamical variability described above. Regarding the role of the choice of reanalysis used for nudging, we find that the $COCl_2$ trends in the two sets of nudged runs are overall similar, albeit with somewhat stronger amplitudes of both the positive and negative responses simulated in the runs nudged to the ERA-Interim reanalysis ($VSLS_{SD-I}$ and $BASE_{SD-I}$).

### 5.3. Individual ensemble members

Unlike for HCl, the $COCl_2$ trends derived from the individual ensemble members of the free-running integrations are largely in agreement with each other. There is still some variability in the magnitudes of the diagnosed trends, in particular in the mid-/ and high latitudes (see Figs. S6 and S7) due to the role of natural interannual variability.



## 6. Summary

Atmospheric impacts of chlorinated very short-lived substances over the recent past (up to and including the year 2019) were assessed using the UM-UKCA chemistry-climate model. Our study constitutes the most up-to-date end-to-end assessment of the impacts of Cl-VSLS. It is also the first published study to examine this topic not only using a CCM but also a whole atmosphere model, thereby simulating surface Cl-VSLS emissions, their tropospheric chemistry and transport, as well as the resulting stratospheric impacts in a fully consistent manner.

First, we quantified the transport of Cl-VSLS into the stratosphere using a small ensemble of free-running simulations. We estimated that the stratospheric source gas injection of chlorine from Cl-VSLS doubled over the first two decades of the 21$^{st}$ century, at ~80 ppt Cl in 2019, compared to ~40 ppt Cl in 2000. Combined with the additional chlorine reaching the stratosphere in the form of product gases, we found that Cl-VSLS provided ~130 ppt Cl to the stratosphere in 2019 (compared to ~70 ppt in 2000). The average 2010-2019 difference of ~100 ppt additional chlorine in the lower/mid-stratosphere in the experiment with Cl-VSLS accounted for ~3 % of total stratospheric chlorine for that period. Thereby, it constituted a small but nonetheless important and growing contribution to the overall chlorine budget that is moreover changing in the opposite sense (i.e. increases in time) to the contribution from the long-lived ODSs.

The choice between a free-running versus nudged model set-up was found to have important consequences for the diagnosed atmospheric Cl-VSLS response. First, the use of the nudged set-up significantly increased the abundance of Cl-VSLS in the lower stratosphere relative to the analogous free-running simulations. Averaged over 2010-2018, the simulation nudged to the ERAI-Interim reanalysis had up to ~20 ppt (i.e. a factor of two) more Cl in the lower stratosphere in the form of Cl-VSLS source gases compared to the free-running case. This arose because of a markedly faster large-scale atmospheric circulation simulated in the nudged run. In comparison, the run nudged to ERA5 showed up to ~10 ppt (i.e. ~50%) more Cl in the form of Cl-VSLS source gases than the free-running case, commensurate with smaller differences in the diagnosed transport between the two. Secondly, despite larger abundance of Cl-VSLS source gases simulated in the lower stratosphere in the nudged runs than in the free-running ones, the corresponding changes in total stratospheric chlorine were slightly smaller. This also related to the faster circulation and indicated reduced concentrations of product gases (e.g. HCl), as opposed to source gases, as well as faster removal of inorganic chlorine from the stratosphere. Regarding the model transport itself, we found that the age-or-air in the simulations nudged to the ERA-Interim reanalysis compared better with that derived from the MIPAS satellite data in the tropics than for the other simulations; however, this was not the case in the mid- and high latitudes.

Our results thus illustrate not only the strong dependence of the diagnosed Cl-VSLS response on the choice between the free-running and nudged model set-up but also on the choice of reanalysis dataset used for nudging. Given that the impact of

nudging on the large-scale residual circulation does not necessarily improve its representation in CCMs (Orbe, et. al., 2018; Chrysanthou et al., 2019), our results illustrate explicitly the first order impact this can have on simulated atmospheric tracers and their responses to external perturbations.

We also investigated the impact of the growth in the atmospheric Cl-VSLS concentrations on the early 21$^{st}$ century HCl and COCl$_2$ trends. We found that the inclusion of Cl-VSLS significantly reduced the magnitudes of the negative HCl and COCl$_2$ trends in the stratosphere (e.g. from ~-8 %(HCl)/decade and ~ -4 ppt(COCl$_2$)/decade at ~20 km to ~-6 %(HCl)/decade and ~ 2 ppt(COCl$_2$)/decade in the free running simulations) , as well as led to positive trends in these tracers in the troposphere. In the tropics, both free-running and nudged integrations with Cl-VSLS included compared better to the observed trends than analogous simulations without Cl-VSLS. Unlike the nudged runs, the ensemble mean free-running integrations did not reproduce the hemispheric asymmetry in the observed mid-latitude HCl and COCl$_2$ trends related to short-term dynamical variability over the recent period. Thus, while accurately reflecting an average Cl-VSLS response under 'mean' dynamical conditions, the ensemble mean free-running integrations may be less useful for direct comparison with atmospheric observations (which effectively constitute only one ensemble member). Indeed, a comparison of HCl trends derived from the individual ensemble members of the free-running integrations revealed a considerable spread of the diagnosed tracer trends, illustrating the role of natural interannual variability in modulating the diagnosed responses. Our results highlight the need for caution when interpreting both model and observed tracer trends derived over a relatively short time period (here ~15 years).

The results in this paper constitute Part 1 of a three-part study of atmospheric impacts of Cl-VSLS. In the follow up to this work, we will focus on the impacts of Cl-VSLS on stratospheric ozone (Part 2), and compare simulations forced with lower boundary conditions of Cl-VSLS, as used here, with simulations using recently estimated Cl-VSLS emissions from Claxton et al. (2020) (Part 3).

**Acknowledgements**

EMB was supported by the UK Natural Environment Research Council (NERC) SISLAC project (NE/R001782/1). RH was supported by the NERC Independent Research Fellowship (NE/N014375/1), the NERC ISHOC project (NE/R004927/1), and the NERC SISLAC project (NE/R001782/1).
The simulations were carried out using MONSOON2, a collaborative High-Performance Computing facility funded by the Met Office and the Natural Environment Research Council, and using the ARCHER2 UK National Supercomputing Service.



**Author contributions**

EMB performed the simulations, analysed the results and wrote the first draft of the manuscript. RH, EMB and MPC
designed the study. EMB performed UM-UKCA chemistry scheme developments, with technical guidance from NLA and
scientific guidance from RH. All authors contributed to the discussion of the results and writing of the manuscript.

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

| | BASE | BASE$_{SD-5}$ | BASE$_{SD-I}$ | VSLS | VSLS$_{SD-5}$ | VSLS$_{SD-I}$ |
|---|---|---|---|---|---|---|
| **No of ensemble members** | 3 | 1 | 1 | 3 | 1 | 1 |
| **Length** | 01/1990 - 12/2019 | 01/1990 - 03/2020 | 01/1990 – 08/2019 | 01/1990 - 12/2019 | 01/1990 - 03/2020 | 01/1990 – 08/2019 |
| **Meteorology (T, u, v)** | free-running | nudged to ERA5 | nudged to ERA-Interim | free-running | nudged to ERA5 | nudged to ERA-Interim |
| **SSTs/sea-ice** | monthly mean* | daily mean** | daily mean** | monthly mean* | daily mean** | daily mean** |
| **Cl-VSLS** | none | none | none | LBCs | LBCs | LBCs |

**Table 1. Summary of UM-UKCA integrations performed. * Durack and Taylor (2016) over 1990-2016, Reynolds and Smith (1994) over 2017-2019. ** Reynold and Smith (1994).**


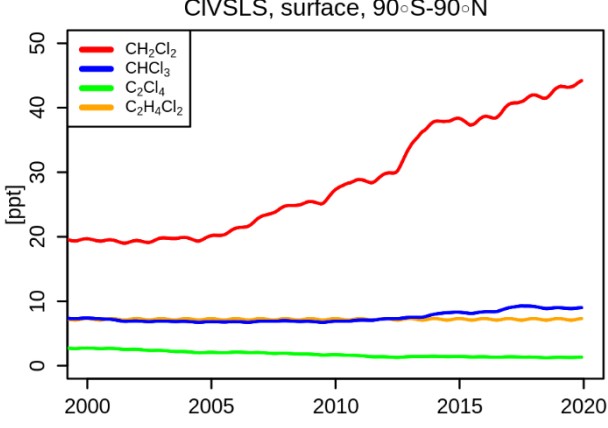

**Figure 1. Global mean volume mixing ratios [ppt] of CH₂Cl₂ (red), CHCl₃ (blue), C₂Cl₄ (green) and C₂H₄Cl₂ (orange) simulated at the surface in the VSLS experiments.**

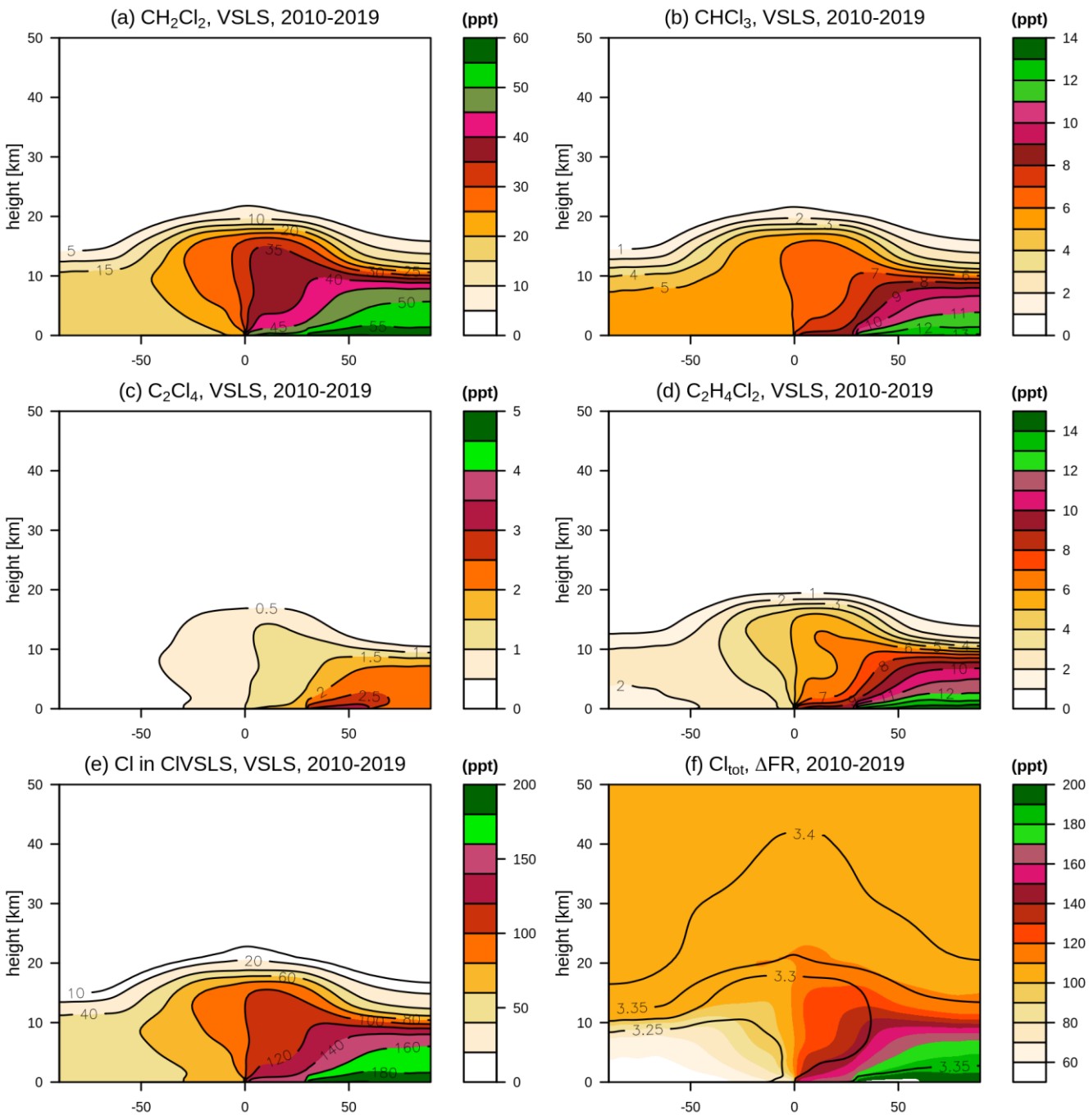

Figure 2. Annual mean zonal mean 2010-2019 volume mixing ratios [ppt] of: (a) CH₂Cl₂, (b) CHCl₃, (c) C₂Cl₄, (d) C₂H₄Cl₂, and (e) Cl in Cl-VSLSs source gases simulated in the ensemble mean VSLS experiments. Panel (f) shows the corresponding difference in Cl$_{tot}$ [ppt] between the ensemble mean VSLS and BASE (i.e. the ΔFR response); contours show Cl$_{tot}$ [ppb] in the VSLS experiment for reference.





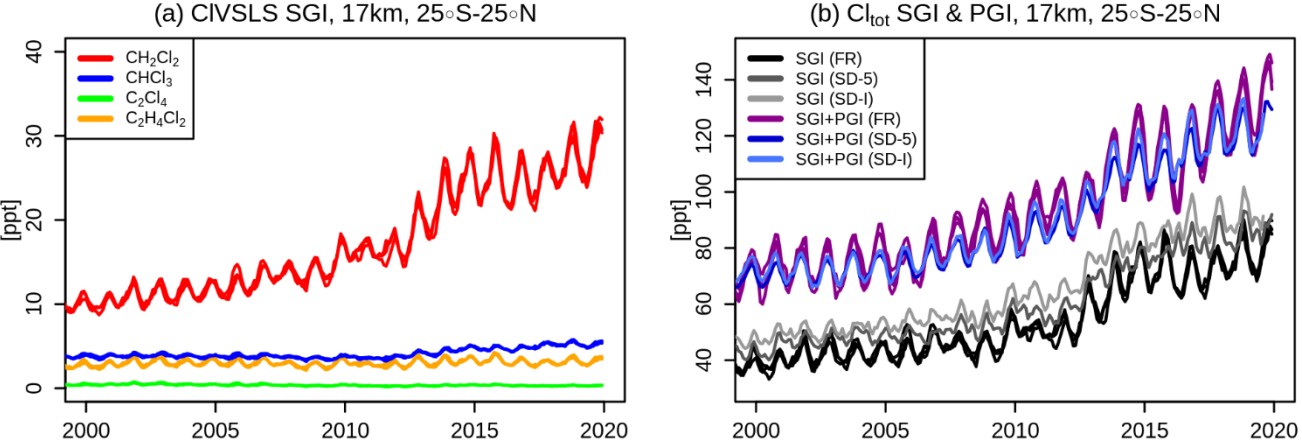

**Figure 3. Monthly mean volume missing ratios [ppt] at 25°S-25°N and 17 km for (a) CH₂Cl₂ (red), CHCl₃ (blue), C₂Cl₄ (green) and C₂H₄Cl₂ (orange) in the free-running VSLS experiments, and (b) Cl present in the Cl-VSLS (i.e. SGI, black) in the VSLS experiments, and the difference in Cl_tot between the VSLS runs and the ensemble mean BASE (i.e. SGI+PGI, purple). Grey lines in (b) indicate SGI in the nudged VSLS_SD-5 and VSLS_SD-I runs, and blue lines the respective SGI.**

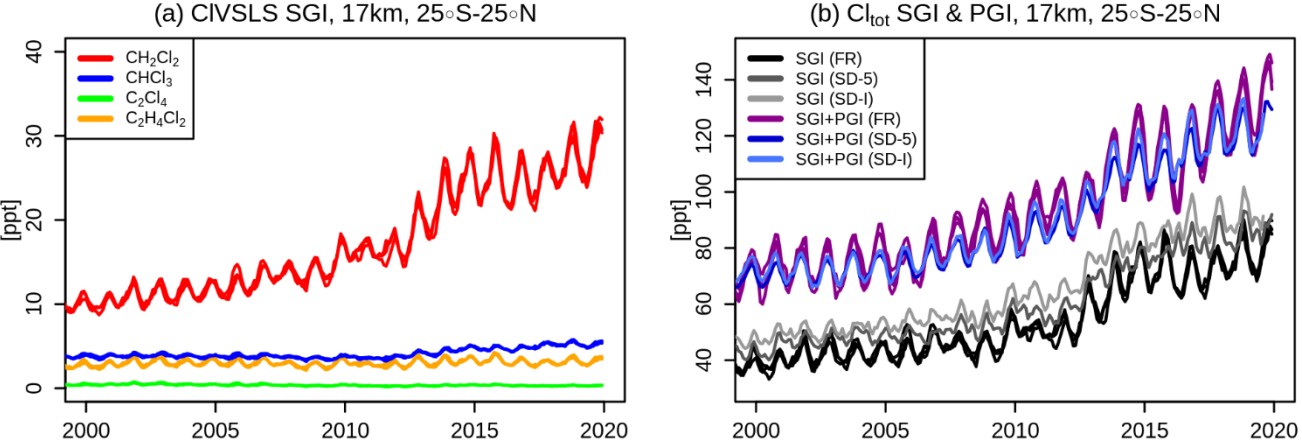


**Figure 4. Shading: Annual and zonal mean 2010-2019 difference [ppt] in: (a) HCl, (b) ClONO₂, (c) COCl₂, and (d) ClO between the ensemble mean VSLS and BASE experiments (i.e. the ΔFR responses). Contours show the corresponding values in the ensemble mean VSLS for reference; note that contours in (a,b) are in ppb.**




**Figure 5. Shading: 2010-2018 difference in Cl [ppt] in: (a,c) Cl-VSLSs source gases between (a) VSLS$_{SD-5}$ and the ensemble mean VSLS, and between (c) VSLS$_{SD-I}$ and the ensemble mean VSLS; (b, d) the difference between the Cl$_{tot}$ responses in (b) ΔSD-5 and**
**ΔFR and in (d) ΔSD-I and ΔFR. Contours show the corresponding values in the free running VSLS for reference.**





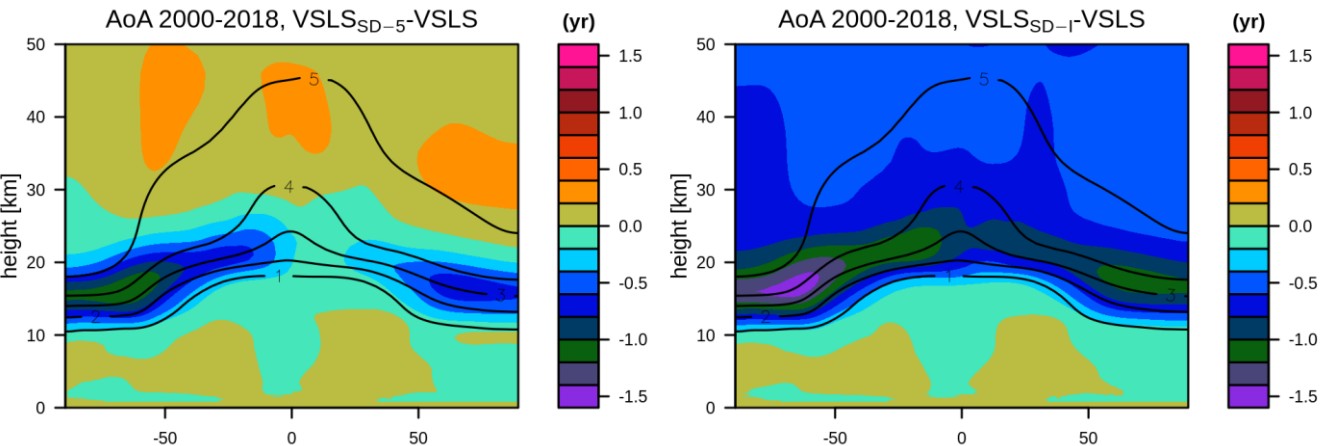

**Figure 6. Shading: 2010-2018 difference in the model age-of-air [yr] between the nudged and free-running integrations. (a) The difference between VSLS$_{SD-5}$ and the ensemble mean VSLS, and (b) the difference between VSLS$_{SD-I}$ and the ensemble mean VSLS. Contours show the age-of-air in the ensemble mean free-running VSLS for reference.**





**Figure 7. Normalised age-of-air [yr] for (a) 25°S-25°N, (b) 30°S-60°N, (c) 30°N-60°S and (d) at 21 km, diagnosed from the free-running VSLS (red) and the nudged VSLS$_{SD-5}$ (green) and VSLS$_{SD-I}$ (blue) simulations. Black shows the corresponding AoA derived from the MIPAS SF6 satellite observations (black; Stiller et al., 2020). Both model and observed AoA were averaged over the 7-year period from May 2005 to Apr 2012 inclusive. Both model and observed AoA were normalised to be zero at the tropical tropopause by subtracting the values calculated in each case for the tropical tropopause layer (here approximated as mean over 25°S-25°N, 16-17 km).**






**Figure 8. Linear trends in deseasonalised HCl mixing ratios [%/10yr] in (a) the ensemble mean VSLS, (b) ensemble mean BASE, (c) the nudged VSLS$_{SD-5}$, (d) BASE$_{SD-5}$, (e) VSLS$_{SD-I}$, (f ) BASE$_{SD-I}$, and from (g) the observed ACE-FTS satellite data (g). Trends in (a, b, c, d, g) are calculated over MAM2004 – SON2019; trends in (e,f) are calculated over MAM2004 – JJA2019. Hatching indicates statistical significance, here taken as regions where the magnitude of the derived trend exceeds ±2 standard errors.**






**Figure 9. Linear trends in deseasonalised HCl mixing ratios [%/10yrs] averaged over (top) 30°N-60°N, (middle) 30°S-30°N and (bottom) 30°S-60°S. Left to right panels in each row are for: (a,e,i) the ensemble mean VSLS (red), ensemble mean BASE (black) and ACE-FTS (green); (b,f,j) VSLS_SD-5 (red), BASE_SD-5 (black) and ACE-FTS (green); (c,g,k) VSLS_SD-I (red), BASE_SD-I (black) and ACE-FTS (green); (d,h,l) the individual VSLS ensemble members (red) and the individual BASE ensemble members (black). Errorbars indicate confidence intervals (±2 standard errors); errorbars are not plotted in (d,h,l) for clarity.**






**Figure 10.** As in Fig. 8(a-f) but for trends in deseasonalised model AoA [day/10yrs].





**Figure 11. Linear MAM 2004 – SON 2019 trends in deseasonalised HCl mixing ratios [%/10yr] derived from the individual VSLS ensemble members: (a) ENS1 VSLS, (c) ENS2 VSLS, and (e) ENS3 VSLS, as well as from the individual BASE ensemble members: (b) ENS1 BASE, (d) ENS2 BASE, (f) ENS3 BASE. Hatching indicates statistical significance (as in Fig. 6).**






Figure 12. As in Fig. 8 but for linear trends in deseasonalised COCl$_2$ mixing ratios [ppt/10yrs].