# Peer review of "Atmospheric impacts of chlorinated very short-lived substances over the recent past. Part 1: stratospheric chlorine budget and the role of transport"

_Atmospheric Chemistry and Physics, 2022_

## Referee Comment (RC1)

**Review of the manuscript "Atmospheric impacts of chlorinated very short-lived substances over the recent past. Part 1: the role of transport" by Bednarz et al., ACPD, 2022.**

The paper presents a modeling study using a whole atmosphere CCM to evaluate how sensitive is the stratospheric loading of chlorinated very short-lived species (Cl-VSLS), including both Source Gas Injection (SGI) and Product Gas Injection (PGI), to different model configurations for the dynamical transport (i.e., considering free-running and nudged setups). The analysis focused on the CL-VSLS evolution during the 2000-2019 period, where they found an overall SGI enhancement from 40 ppt to 80 ppt for the free-running simulation, and up to 10-20 ppt additional Cl-VSL when the model was nudged to ERA-Interim or ERA-5 reanalysis. The larger SGI for the nudged configurations resulted in an overall smaller SGI + PGI due to the faster transport. To evaluate the total inorganic chlorine evolution in the stratosphere, they present a comparison of HCl and $COCl_2$ trends in the stratosphere and upper troposphere with satellite observations (ACE-FTS). They show that regardless of the transport configuration, the inclusion of Cl-VSLS improves the model performance, although the hemispheric asymmetry observed in the lower stratosphere is only captured with the nudged simulations. The work is very well planed and provides a realistic and clear evaluation of the magnitude of the Cl-VSLS contribution to the total inorganic chlorine loading in the lower stratosphere. The methodology and results are generally well presented, although some clarification is required as described below. I suggest the paper is accepted for publication after the following issues have been solved:

**Main Comments:**

**1. Splitting the project in Part 1, Part 2, Part 3.**
The authors decided to split the paper in 3 parts, but what each of the parts is about is only clear in the last sentence of the conclusions (P12,L371-375). I suggest making it clear since the beginning to avoid the reader wondering him/her-self about what the other parts will address. Note that in P6,L186, the authors explicitly mention that the HCl comparison with satellite data will be presented in a following paper, but it is included here in this draft (Section 4).

**2. Year-to-Year variability for individual free-running ensemble members**
I found it very surprising the very small variability on the SGI (as well as for the SGI + PGI) among the 3 individual members of the free-running ensemble (Fig. 3). I wonder how large is the year-to-year variability for the free-running simulations, in comparison with the difference between the free-running mean and each of the nudged simulations? For example, during the

first 5 years after the spin-up (when the Cl-VSLS LBCs remain almost constant), the year-to-year variability for a single ensemble is much larger than the variability between the free-running individual simulations during a single year. Then, could this year to year variability in the free-running simulations be more representative of the model variability than computing the multi-ensemble mean?

In addition, when reading section 4.3 (Fig. 9, rightmost pannels) it is surprising that HCl trends for the different ensemble members present such a large variability, while the Cl-VSLS variability between ensembles from Fig. 3 seems to be very small. Have you computed the Cl-VSLS trends using the same procedure (and in the same units as for the HCl trends, % per decade) to evaluate if the SGs are also showing a large range of trends for each ensemble-member as for the PGs?

**3. The role of Chemistry representation**

The impact of transport on the total SGI is explained in detail. However, the sum of SGI + PGI is only briefly discussed in Section 3 (P7,219-224). In particular, the individual PGI contribution is not addressed for any of the configurations, and no individual number is given. It is only mentioned that the trends and variability of the SGI + PGI are smaller for the nudged simulations than for the free-running. Even though I understand the authors focus the product gas discussion on Section 4 and 5 where they compare with HCl and COCl$_2$ observations, I believe the paper would benefit of extending a bit the discussion of the overall PGI before moving on the Age of Air trends. Unless the PGI is going to be presented in detail in Part 2 of the paper. In addition to i) the enhanced transport for the nudged simulations (P6,L222) and ii) the faster large scale circulation in the stratosphere (P6,L223), the change in PGs abundance in the upper TTL can affect the washout efficiency of halogenated species and therefore modify the overall PGI (see Fernandez et al., 2021 for the case of bromine).

**Minor Comments:**

P5,L160-162: What is the mean value for SGI for the final year 2019? Because by looking at Fig. 3 it looks smaller than 80 ppt for the annual mean. The text mentions a couple of times that the Cl-VSLS is "doubled" … just wanted to be sure if is not "almost doubled".

P7,L227-P8,L233: It is not clear how the seasonal mean for MAM and SON (and the other seasons, none of them are defined) was computed. Furthermore, the text mention that the

model output was deseasonalised and then explain that the seasonal mean was used. I suggest re-order and re-phrase to make it clear.

P8,L249: "a strong asymmetry in its horizontal structure". Do you mean a hemispheric asymmetry?

P8,L259-P9,265: Section 4.2 begins mentioning that "nudged simulations … show similar interanual dynamical variability to observations", and later concludes that "such pattern is thus similar to that found in the ACE-FTS data". The analysis is correct, but first, it should be "more" similar (not just similar), and second it is not clear if the initial sentence points at the AoA comparison until it is later explained in the paragraph. I suggest re-ordering.

P10,L317: Any explanation on why $COCl_2$ shows a much smaller variability than HCl for the same set of ensembles?

P11,L345-346: It should be made clear that the sentence applies for the nudged simulations.

P12, L366: What do you mean by "which effectively constitute only one ensemble member"?

**Language editing comments and Typos:**

P2,L58: Engel and Rigby et al., 2018 (here and elsewhere).

P3,L88; P4,L118-120, P8,L235: Consecutive points and/or double-spaces.

P5,L159-160: "The stratospheric source gas injection ('SGI') of chlorine from Cl-VSLS can be approximated based on their simulated concentrations at 17 km and 25°S-25°N". The term can be approximated here can introduce confusion as someone can infer that you did not computed it this way. I suggest to rephrase.

P6,L166-170: The sentence is confusing: at the beginning you said that "experiments have, by definition, different meteorology". But later it is mentioned that "ensemble members are forced with identical chemical (Cl-VSLS) and meteorological (SST and sea-ice) LBCs". I suggest using oceanic/surface LBCs at the end of the sentence.

P6,L185: What do you mean by "brought about from the inclusion"?

P6, L191: Move "more Cl" before the opening parenthesis to make it clear what the 20 ppt are.

P8,L229: Define ACE-FTS

P8,L251: Fig. 7g →Fig. 8g.

P9,L264: Fig. 7c-f →Fig. 8c-f.

P9,L270-274: Make sure you point at the proper panels of Fig. 9 for the tropics, NH and SH regions.

P10,L300;P10,L311: Fig. S6 → Fig. S7.

P11,L334: Is it "sense" or "sign" ?

P18,L568: Remove Sturges et al., 2000 at the end of the reference.

**Figures and Tables**

Figure 1: The figure shows the Cl-VSLS surface LBCs. I understand this is an "input" to force the model at the surface and not an "output" of the model. If that is the case then instead of "simulates" it could be changed to "forced" or something similar this.

Figure 2: "Annual mean zonal mean" reads awkward. Also note that Cl/VSLS in panel e and Cl_tot in panel f are not defined.

Figure 3 (also Fig. 7 and elsewhere). Note that the degree symbol is printed erroneously on the pdf file.

Figure 3 caption: At the very end of the caption it should be "SGI + PGI". Make it explicit in the caption that the 3 individual members of the free-running ensemble are shown.

Figure 10: I found it surprising that AoA trends for panels a and b showed such a different distribution. Does it impacts on the analysis? Should it be highlighted in the text?

**Supplement:**

Affiliations are missing. They should be made consistent with the main text.

Fig. S1: missing →mixing

Fig. S4: It should be made clear in the caption that results are for each ensemble

Fig. S6: main paper → main text.

**References**

Engel, A., M. Rigby, J.B. Burkholder, R.P. Fernandez, L. Froidevaux, B.D. Hall, R. Hossaini, T. Saito, M.K. Vollmer, and B. Yao, Update on Ozone-Depleting Substances (ODSs) and Other Gases of Interest to the Montreal Protocol, Chapter 1 in Scientific Assessment of Ozone Depletion: 2018, Global Ozone Research and Monitoring Project–Report No. 58, World Meteorological Organization, Geneva, Switzerland, 2018.

Fernandez, R.P., J.A. Barrera, A.I. López-Noreña, D.E. Kinnison, J. Nicely, R.J. Salawitch, P.A. Wales, B.M. Toselli, S. Tilmes, J.-F. Lamarque, C.A. Cuevas, and A. Saiz-Lopez, Intercomparison between surrogate, explicit and full treatments of VSL bromine chemistry within the CAM-Chem chemistry-climate model, Geophys. Res. Lett., 48(4), doi:10.1029/2020GL091125, 2021.

---

## Author Comment (AC1)

We thank both reviewers for positive reviews as well as helpful comments for improving the manuscript. We address all comments in blue below.

**REVIEW #1**

Review of the manuscript "Atmospheric impacts of chlorinated very short-lived substances over the recent past. Part 1: the role of transport" by Bednarz et al., ACPD, 2022.

The paper presents a modeling study using a whole atmosphere CCM to evaluate how sensitive is the stratospheric loading of chlorinated very short-lived species (Cl-VSLS), including both Source Gas Injection (SGI) and Product Gas Injection (PGI), to different model configurations for the dynamical transport (i.e., considering free-running and nudged setups). The analysis focused on the CL-VSLS evolution during the 2000-2019 period, where they found an overall SGI enhancement from 40 ppt to 80 ppt for the free-running simulation, and up to 10-20 ppt additional Cl-VSL when the model was nudged to ERA-Interim or ERA-5 reanalysis. The larger SGI for the nudged configurations resulted in an overall smaller SGI + PGI due to the faster transport. To evaluate the total inorganic chlorine evolution in the stratosphere, they present a comparison of HCl and COCl2 trends in the stratosphere and upper troposphere with satellite observations (ACE-FTS). They show that regardless of the transport configuration, the inclusion of Cl-VSLS improves the model performance, although the hemispheric asymmetry observed in the lower stratosphere is only captured with the nudged simulations. The work is very well planed and provides a realistic and clear evaluation of the magnitude of the Cl-VSLS contribution to the total inorganic chlorine loading in the lower stratosphere. The methodology and results are generally well presented, although some clarification is required as described below. I suggest the paper is accepted for publication after the following issues have been solved:

Main Comments:

1. Splitting the project in Part 1, Part 2, Part 3. The authors decided to split the paper in 3 parts, but what each of the parts is about is only clear in the last sentence of the conclusions (P12,L371-375). I suggest making it clear since the beginning to avoid the reader wondering him/her-self about what the other parts will address. Note that in P6,L186, the authors explicitly mention that the HCl comparison with satellite data will be presented in a following paper, but it is included here in this draft (Section 4).

We agree with the reviewer and have changed the manuscript to clarify this. In particular, we have changed the title to "… Part 1. stratospheric chlorine budget and the role of transport". We have also changed the sentence "Atmospheric impacts of chlorinated very short-lived substances …" in the first sentence of both the abstract and summary to "Impacts of chlorinated very short-lived substances on stratospheric chlorine budget …". We have also included the information on what future papers (Part 2 and 3) will address in the last paragraph of the introduction.

Regarding the information in P6 L186, this refers to the comparison of climatological HCl concentrations with satellite data, which will be discussed in the parallel DEST description paper that is currently under preparation (unlike the comparison of HCl trends that is presented in Section 4 of this manuscript).

2. Year-to-Year variability for individual free-running ensemble members I found it very surprising the very small variability on the SGI (as well as for the SGI + PGI) among the 3 individual members of the free-running ensemble (Fig. 3). I wonder how large is the year-to-year variability for the free-running simulations, in comparison with the difference between the free-running mean and each of the nudged simulations? For example, during the first 5 years after the spin-up (when the Cl-VSLS LBCs remain almost constant), the year-to-year variability for a single ensemble is much larger than the variability between the free-running individual simulations during a single year. Then, could this year to year variability in the free-running simulations be more representative of the model variability than computing the multi-ensemble mean?

The reviewer is correct that the variability between the individual ensemble members of the free-running experiments is much smaller than the interannual variability in each of the experiments itself. As we discuss in the second paragraph of Section 3.1, this suggests that it's the interannual differences in prescribed Cl-VSLS LBCs and SSTs/sea-ice rather than the model internal dynamical variability that is the main driver of variability in SGI and PGI on interannual timescales. We now expand this by explicitly referring to Fig. S2 as an illustration of variability in tropical SSTs, and note that 'interannual variability in tropical SSTs, in particular, has been previously shown to be an important driver of variability in large scale atmospheric circulation (e.g. Neu et al., 2014); this effect will thus likely play an important role for modulating the stratospheric Cl-VSLS transport.'

In addition, when reading section 4.3 (Fig. 9, rightmost pannels) it is surprising that HCl trends for the different ensemble members present such a large variability, while the Cl-VSLS variability between ensembles from Fig. 3 seems to be very small. Have you computed the Cl-VSLS trends using the same procedure (and in the same units as for the HCl trends, % per decade) to evaluate if the SGs are also showing a large range of trends for each ensemble-member as for the PGs?

We believe that the variability in the HCl trends inferred from the individual ensemble members results primarily from the background natural variability in atmospheric circulation (as illustrated by the differences in AoA trends inferred from the individual ensemble members) and, thus, HCl concentrations, which can lead to uncertainty in the inferred HCl trends. We now try to further explain and clarify this in Section 4.3. The impacts of variability in the Cl-VSLS transport (caused also by the background natural variability in atmospheric circulation) on the inferred HCl trends is likely of secondary importance, as indicated by the much smaller differences in SGI+PGI between the individual ensemble members (as pointed by the reviewer).

3. The role of Chemistry representation The impact of transport on the total SGI is explained in detail. However, the sum of SGI + PGI is only briefly discussed in Section 3 (P7,219-224). In particular, the individual PGI contribution is not addressed for any of the configurations, and no individual number is given. It is only mentioned that the trends and variability of the SGI + PGI are smaller for the nudged simulations than for the free-running. Even though I understand the authors focus the product gas discussion on Section 4 and 5 where they compare with HCl and COCl2 observations, I believe the paper would benefit of extending a bit the discussion of the overall PGI before moving on the Age of Air trends. Unless the PGI is going to be presented in detail in Part 2 of the paper. In addition to i) the enhanced transport for the nudged simulations (P6,L222) and ii) the faster large scale circulation in the stratosphere (P6,L223), the change in PGs abundance in the upper TTL can affect the washout efficiency of halogenated species and therefore modify the overall PGI (see Fernandez et al., 2021 for the case of bromine).

We thank the reviewer for these helpful suggestions. We have now added more discussion about the PGI to the last paragraph in Section 3.1. In particular, in addition to the sentences:

"Figure 4 speciates the modelled difference in chlorine between the experiments VSLS and BASE. Most of the additional Cl in the former runs is found as HCl, the principal stratospheric chlorine reservoir, and in ClONO2, ClO and COCl2 (phosgene). The latter is an important product of Cl-VSLS oxidation and an atmospheric degradation product of the longer-lived source gases CCl4 and CH3CCl3 (e.g. Fu et al., 2007; Harrison et al., 2019). Here, we estimate that up to 8 ppt of the 195 COCl2 simulated over the last decade in the VSLS experiment is of Cl-VSLS origin, with Cl-VSLS accounting for the majority of COCl2 found in the troposphere"

that were already in Section 3.1, we now added:

'In terms of the contribution to the stratospheric chlorine injection from product gases (PGI), we find that averaged over 2010-2019 the inclusion of Cl-VSLS results in 27 ppt more Cl injected in the form of HCl and 12 ppt more Cl injected in the form of COCl2 compared to BASE. This can be compared to the 66 ppt Cl injected as Cl-VSLS source gases on average over the same period.'

We have also rephrased the last paragraph of Section 3.2 to specify that both the increase in PGI and the increase in the stratospheric levels of PG are smaller in nudged UM-UKCA runs than in the free running ones. We also mention the suggested changes in washout efficiency as a possible contributor to the simulated responses.

Minor Comments:

P5,L160-162: What is the mean value for SGI for the final year 2019? Because by looking at Fig. 3 it looks smaller than 80 ppt for the annual mean. The text mentions a couple of times that the Cl-VSLS is "doubled" … just wanted to be sure if is not "almost doubled".

The annual mean free-running ensemble mean SGI is 38.8 ppt in 2000 and 78.5 ppt in 2019. Hence we believe it is correct to say in the text that 'the simulated SGI doubled over the first two decades of the 21st century, with ~80 ppt Cl being injected into the stratosphere in the form of Cl-VSLS in 2019, compared to ~40 ppt Cl in 2000'.

P7,L227-P8,L233: It is not clear how the seasonal mean for MAM and SON (and the other seasons, none of them are defined) was computed. Furthermore, the text mention that the model output was deseasonalised and then explain that the seasonal mean was used. I suggest re-order and re-phrase to make it clear.

We have now clarified this in the text.

P8,L249: "a strong asymmetry in its horizontal structure". Do you mean a hemispheric asymmetry?

Yes, we have now clarified this in the text.

P8,L259-P9,265: Section 4.2 begins mentioning that "nudged simulations … show similar interanual dynamical variability to observations", and later concludes that "such pattern is thus similar to that found in the ACE-FTS data". The analysis is correct, but first, it should be "more" similar (not just similar), and second it is not clear if the initial sentence points at the AoA comparison until it is later explained in the paragraph. I suggest re-ordering.

As suggested, we have changed 'such pattern is thus similar' to 'such pattern is thus more similar'. We note that the phrase 'nudged simulations … show similar interannual dynamical variability to observations' in the first sentence is a general statement. A particular illustration of this in the form

of age-of-air trends is made only in the second sentence, where the reference to Fig. 10 is made. We apologise for the confusion.

P10,L317: Any explanation on why COCl2 shows a much smaller variability than HCl for the same set of ensembles?

Unfortunately, the reasons behind this behaviour are not clear to us.

P11,L345-346: It should be made clear that the sentence applies for the nudged simulations.

We have now clarified that.

P12, L366: What do you mean by "which effectively constitute only one ensemble member"?

We mean that the real world effectively constitutes only one ensemble member, i.e. possible realization. We have now clarified this in the text.

Language editing comments and Typos:

P2,L58: Engel and Rigby et al., 2018 (here and elsewhere).

Corrected

P3,L88; P4,L118-120, P8,L235: Consecutive points and/or double-spaces.

Corrected

P5,L159-160: "The stratospheric source gas injection ('SGI') of chlorine from Cl-VSLS can be approximated based on their simulated concentrations at 17 km and 25°S-25°N". The term can be approximated here can introduce confusion as someone can infer that you did not computed it this way. I suggest to rephrase.

Changed to 'calculated'.

P6,L166-170: The sentence is confusing: at the beginning you said that "experiments have, by definition, different meteorology". But later it is mentioned that "ensemble members are forced with identical chemical (Cl-VSLS) and meteorological (SST and sea-ice) LBCs". I suggest using oceanic/surface LBCs at the end of the sentence.

Changed to 'oceanic'

P6,L185: What do you mean by "brought about from the inclusion"?

We now specify 'brought about from the inclusion of Cl-VSLS LBCs and chemistry'

P6, L191: Move "more Cl" before the opening parenthesis to make it clear what the 20 ppt are.

Corrected

P8,L229: Define ACE-FTS

Corrected

P8,L251: Fig. 7g ☐Fig. 8g.

Corrected

P9,L264: Fig. 7c-f ⬚Fig. 8c-f.

Corrected

P9,L270-274: Make sure you point at the proper panels of Fig. 9 for the tropics, NH and SH regions.

Corrected

P10,L300;P10,L311: Fig. S6 ⬚ Fig. S7.

Corrected

P11,L334: Is it "sense" or "sign" ?

'the opposite sense'

P18,L568: Remove Sturges et al., 2000 at the end of the reference.

Corrected

Figures and Tables

Figure 1: The figure shows the Cl-VSLS surface LBCs. I understand this is an "input" to force the model at the surface and not an "output" of the model. If that is the case then instead of "simulates" it could be changed to "forced" or something similar this.

The figure shows an 'output' and so the word 'simulates' is correct. We note that these should be very close to values 'input' to the model.

Figure 2: "Annual mean zonal mean" reads awkward. Also note that Cl/VSLS in panel e and Cl_tot in panel f are not defined.

We've now corrected ClVSLS to Cl-VSLS in panel e, and define $Cl_{tot}$ in the figure caption.

Figure 3 (also Fig. 7 and elsewhere). Note that the degree symbol is printed erroneously on the pdf file.

Figure 3 caption: At the very end of the caption it should be "SGI + PGI". Make it explicit in the caption that the 3 individual members of the free-running ensemble are shown.

Corrected

Figure 10: I found it surprising that AoA trends for panels a and b showed such a different distribution. Does it impacts on the analysis? Should it be highlighted in the text?

We thank the reviewer for careful consideration of our results. We agree that there are differences in the AoA trends derived from the ensemble mean VSLS and BASE experiments (panels a and b). We note that these difference are much smaller than that the differences in AoA trends diagnosed from the individual ensemble members (Fig. S5 of the Supplement), which is the point we want to stress in the text in the first instance (and we do that in Section 4.3) as this has a strong impact on the HCl trends derived from the individual ensemble members shown in Fig. 11.

Supplement:

Affiliations are missing. They should be made consistent with the main text.

Corrected

Fig. S1: missing ￼mixing

Corrected

Fig. S4: It should be made clear in the caption that results are for each ensemble

Corrected

Fig. S6: main paper ￼ main text.

Corrected

We thank both reviewers for positive reviews as well as helpful comments for improving the manuscript. We address all comments in blue below.

**REVIEW #2**

Bednarz et al. present a study exploring the transport of Cl-VSLS into the stratosphere in both free running and nudged version of their chemistry-climate model. I found this study to be very interesting, exploring an important topic in the contribution of Cl-VSLS to recent stratospheric chlorine changes. The comparison of the free running and nudged simulations nicely demonstrates the sensitivity of Cl-VSLS injection to model dynamical fields. The paper is clearly written and laid out and the figures are of high quality. However, while I found the analysis presented in the paper to be fine, the framing of that analysis and the conclusions reached in the paper should be tempered, particularly the conclusions regarding the sensitivity to choice between free running and nudged model configurations. I also feel that as the study is part of three planned manuscripts, more should be said earlier in this manuscript about what will be covered here and what elsewhere. The title of this manuscript implies that wider atmospheric impacts will be evaluated, and it is only clear in the final paragraph that changes to ozone will be considered in a follow-up. I would recommend publication of the manuscript after the authors address the comments below.

General comments:

The paper is titled atmospheric impacts of chlorinated VSLS, but there is very little evaluation of the impacts of Cl-VSLS on wider atmospheric chemistry/chemistry-climate coupling in the paper. Instead, the paper is really an evaluation of the impact of Cl-VSLS on stratospheric chlorine, particularly HCl. This is probably because the authors plan to publish their work in three parts. But it is only made clear at the end of the manuscript what will be included in the other parts. The authors should move this information to much earlier in the manuscript. I also feel sentences like 'Atmospheric impacts of chlorinated very short-lived substances over the recent past (up to and including the year 2019) were assessed using the UM-UKCA chemistry-climate model', as used in L321-322 should be qualified by making it clear the authors in this study are only looking at the impact of VSLS on stratospheric halogens, and the sensitivity of this to model dynamics.

We agree with the reviewer and have changed the manuscript to clarify this. In particular, we have changed the title to "… Part 1. stratospheric chlorine budget and the role of transport". We have also changed the sentence "Atmospheric impacts of chlorinated very short-lived substances …" in the first sentence of both the abstract and summary to "Impacts of chlorinated very short-lived substances on stratospheric chlorine budget …". We have also included the information on what future papers (Part 2 and 3) will address in the last paragraph of the introduction.

An argument that is made by the paper that I'm not sure I agree with is that model results are very sensitive to the choice to run the model in free running or nudged modes, i.e., the model configuration. But presumably this arises because there are biases/differences between the modelled dynamics compared to the reanalysis datasets. If so, I feel the argument should be changed to 'our results highlight the importance of model dynamical fields on transport of Cl-VSLS into the lower stratosphere'. In a model that has dynamical fields that look very similar to the reanalysis datasets, the authors could have concluded that there is very little sensitivity to the choice to run the model in free running or nudged modes. Otherwise, if the authors are arguing that the

process of nudging the model is introducing additional tendencies into the transport of Cl-VSLS species that simply do not exist in the free running version of the model, or that this different configurations fundamentally change composition-dynamical coupling/feedbacks in such as way as to significantly influence the modelling of Cl-VSLS, then I don't feel the authors show enough to conclude this in this study.

We agree with the reviewer and have modified the text.

In particular, in abstract we changed 'The results illustrate the strong dependence of the simulated stratospheric Cl-VSLS levels to the choice between free-running versus nudged set-up, …' to 'The results illustrate the strong dependence of the simulated stratospheric Cl-VSLS levels on the model dynamical fields. In UM-UKCA this corresponds to the choice between free-running versus nudged set-up, …' .

In the summary, we changed 'Our results thus illustrate not only the strong dependence of the diagnosed Cl-VSLS response on the choice between the free-running and nudged model set-up …' to 'Our results highlight the importance of model dynamical fields on transport of Cl-VSLS into the lower stratosphere. In UM-UKCA this corresponds to the strong dependence of the diagnosed stratospheric Cl-VSLS levels on the choice between the free-running and nudged model set-up …'.

We also now stress throughout the main text that the results are model specific.

The DEST chemical scheme used in the paper is cited as in prep. Is this still the case? It seems to me that this scheme is key to producing and interpreting the results presented in the paper, but the reader doesn't know what it is, or how it models stratospheric halogens. If the paper has not yet been published, I would encourage the authors to include much more information on what is in the DEST scheme here.

The reviewer correctly points out that a parallel manuscript including a detailed description and evaluation of the DEST scheme is still under preparation. We have now added more information to Section 2.1 listing all halogenated source gas tracers included in DEST and how their concentrations are forced at the surface. In Section 2.2 we already include a detailed description of the Cl-VSLS LBC timeseries that drive the model. While we acknowledge that some more detailed information about, e.g., specific reactions and inorganic halogen tracers included in DEST would be useful, we would prefer to focus in this paper on scientific results and not to go into too much length regarding detailed aspects of the chemical scheme, which we anticipate will be published separately in near distant future. Before that, we are happy to answer any questions regarding details of the chemical scheme from an interested reader.

Is it possible to add the tropopause to all the latitude height cross sections? The focus on the paper is the transport of Cl-VSLS into the stratosphere, and it would aid the reader to show where that is in this model.

We thank the reviewer for the useful suggestion. We have now added the tropopause to Figure 2.

Specific comments:

L22-31: I found it difficult to follow the arguments in this paragraph. The abstract states that in 2019 there is 80 ppt of stratospheric Cl from source gas injection, and an additional 50 ppt from product gas injection, for a total of 130 ppt. The authors then state that nudging the model to ERA-I meteorology results in a 20 ppt increase in lower stratospheric Cl-VSLS source gases, which is

equivalent to a doubling. I feel this paragraph needs to make it clearer where in the stratosphere these numbers are applicable, and to better relate the values in the free running simulation to those in the nudged simulation. It is clearer in the main text, but on first reading I was comparing a change of 20 ppt to the 130 ppt earlier in the paragraph, and so confused by the doubling comment.

We apologise for the confusion and rephrased the abstract to clarify that the first set of numbers refers to the injection of Cl into the stratosphere in 2019, and the second set of numbers refers to the Cl-VSLS concentrations (in units of Cl) simulated in the tropical lower stratosphere at 20 km (and averaged over 2000-2018).

L39: The effectiveness of the Montreal Protocol is better measured by changes to the gases it controls, e.g. CFC-11 mixing ratios. Trends in HCl are of course affected by this, but also by changes to atmospheric transport (as discussed in this paper) and stratospheric chlorine partitioning as CH4 and N2O etc change.

We agree with the reviewer and now say 'for … aidding the monitoring' as to make it more clear that this method is only complementary to other methods

L48: controls seems too strong here – CO2 and water vapour both play important roles in dictating stratospheric temperatures. Similarly 'controlling' on L50, halogens are of course important, but HOx, NOx and Ox cycles a also key to accurately modelling stratospheric ozone.

We agree and changed the words 'controls' and 'controlling' to 'plays a crucial role in modulating' and 'modulating', respectively.

L49-50: Does ozone not absorb longwave radiation at all altitudes?

We have now corrected the sentence to 'The absorption of longwave radiation by ozone in  the troposphere and lower stratosphere, on the other hand, contributes to the greenhouse effect.'

L169: I do not feel that meteorological LBC is a good description of SSTs and sea ice boundary conditions and would suggest this is changed.

We have now changed this to 'oceanic LBCs'

L189-195: I feel care has to be taken when discussing these values. Given the strong gradients in this region, the way the values are currently presented is misleading, and need further clarification. At the level of the 20 ppt contour in figure 5c, the values have approximately doubled as the authors state, but at the level of the 40 ppt contour, which is within a few km of the 20 ppt contour, the increase is ~50%. Presumably above the 20 ppt contour the percentage increase in Cl-VSLS could be very large.

We have edited the text in Section 3.2 making the analysis specific to a particular region, i.e. 25S-25N mean at 20 km altitude. In particular, the above paragraph now reads 'We find that the use of the nudged model set-up in UM-UKCA significantly increases the abundance of Cl-VSLS in the lower stratosphere (Fig. 5a,c) and hence the stratospheric chlorine SGI from Cl-VSLS (Fig. 3a). For example, in the tropical (25°S-25°N mean) lower stratosphere at 20 km altitude the VSLS$_{SD-I}$ run shows 20 ppt more Cl (39 ppt and 19 ppt in the nudged VSLS$_{SD-I}$ and the free running VSLS, respectively, i.e. a factor of two) in the form of source gases relative to the free-running VSLS simulation (averaged over 2010-2018; Fig. 5c). There is also a strong dependence on the reanalysis used for nudging, with the run nudged to ERA5 (VSLS$_{SD-5}$) showing 10 ppt more Cl in this region (29 ppt and 19 ppt in VSLS$_{SD-5}$ and VSLS, respectively, i.e. ~50%) in the form of source gases than in the free-running experiments instead (Fig. 5a).' We have also changed the relevant parts of Section 6 and the abstract accordingly.

L197: Given the increase seems rather uniform over a range of latitudes, I wonder to what extent might tropopause height changes contribute to the mixing ratio changes seen here? Increased tropopause height may also contribute to the younger age of air seen in Figure 6.

We have looked at the tropopause heights simulated in both free-running and nudged simulations (see Fig. R1, below). As shown in Fig. R1, the use of nudged model set up increases the tropopause height in the tropics and decreases it in the mid and high latitudes compared to the free-running simulations. While these differences in tropopause height could in principle contribute to some of the differences in the simulated responses, we note that: (i) the simulated tropopause heights are very similar in the two nudged runs nudged to either ERA-Interim or ERA-5 (red and blue in Fig. R1) while the associated changes in stratospheric Cl-VSLS concentrations and age-of-air differ substantially between the two runs (see Fig. 5 and 6 of main text); and (ii) in any case it is difficult to diagnose to what extent any tropopause changes are the driver of the atmospheric responses, and to what extent are they the result of and/or are just consistent with the atmospheric responses.

[Figure]

**Figure R1.** Yearly mean tropopause height at a function of latitude averaged over 2010-2018 in ensemble mean VSLS run (black) and in the two nudged VSLS$_{SD5}$ and VSLS$_{SDI}$ runs (blue and red, respectively).

L233: Presumably this is because the ERA-Interim data ends on 31st August? Why not calculate all the trends over the same time period? It seems odd to include the 2019 SON mean in all but one of the panels.

That is correct, the length of VSLS$_{SD-I}$ and BASE$_{SD-I}$ simulations is constrained by the length of ERA-Interim, which ends in August 2019.

We choose to calculate the trends in the other simulations over the period up to SON 2019, i.e. slightly longer than for VSLS$_{SD-I}$ and BASE$_{SD-I}$, in order to maximise the length of the timeseries used for the analysis and, hence, improve the signal-to-noise ratio. We note that since the data are

deseasonalised prior to calculating the trends (as described in the manuscript), an inclusion of one additional season/time point is unlikely to be the primary driver of any differences between the trends calculated from the different experiments.

L299: The authors use the tropical average at earlier points in the paper, but only define it as 30S-30N here. I would move this definition to earlier in the manuscript.

We now include this definition also earlier on in Section 4.1.

L321-325: I feel this paragraph is summing up all three parts of the planned study, two of which are not yet available. Can this study claim to be an end-to-end assessment if there is no discussion of stratospheric ozone changes, impacts on stratospheric temperatures, coupling between the composition and dynamics, and wider atmospheric feedbacks?

We used the word 'end-to-end' here as to say that a continuous timeseries of Cl-VSLS emissions was used in the model experiments, and thus the impacts in both early of the 21 century as well as over some of the more recent years are evaluated consistently in a single modelling study. We have now removed this word to avoid confusion. In addition, we have also changed the first sentence "Atmospheric impacts of chlorinated very short-lived substances …" to "Impacts of chlorinated very short-lived substances on stratospheric chlorine budget …" to make it clear what precisely the paper addresses.

L351-352: Care should be taken here – presumably if the modelled dynamics were more consistent with those in the reanalysis datasets the authors would not make this claim. See my general comment above.

We agree and, as noted above, have changed this to 'Our results highlight the importance of model dynamical fields on transport of Cl-VSLS into the lower stratosphere. In UM-UKCA this corresponds to the strong dependence of the diagnosed stratospheric Cl-VSLS levels on the choice between the free-running and nudged model set-up …'

Technical comments:

L51: Replace 'so far' with 'to date'

Corrected

L61: I would add 'comparatively' before 'few studies' – there is now a large and growing body of literature looking at chlorinated VSLS.

Corrected

L119: 2 fullstops

Corrected

L203: ERAI-Interim -> ERA-Interim. Also L214

Corrected

L204: lager -> larger

Corrected

L206: al -> al.

Corrected